# Effects on Synchronization and Reproductive Efficiency of Delaying the Removal of the Intravaginal Progesterone Device by 24 h in the 5d Co-Synch Protocol in Heifers

**DOI:** 10.3390/ani11030849

**Published:** 2021-03-17

**Authors:** Aitor Fernandez-Novo, Sergio Santos-Lopez, Jose Luis Pesantez-Pacheco, Natividad Pérez-Villalobos, Ana Heras-Molina, Juan Vicente Gonzalez-Martin, Susana Astiz

**Affiliations:** 1Veterinary Department, School of Biomedical and Health Sciences, Universidad Europea de Madrid, C/Tajo s/n, Villaviciosa de Odón, 28670 Madrid, Spain; aitorfn@gmail.com; 2Animal Production Department, Veterinary Faculty, Complutense University of Madrid, 28040 Madrid, Spain; sesantos@ucm.es; 3School of Veterinary Medicine and Zootechnics, Faculty of Agricultural Sciences, University of Cuenca, Avda. Doce de Octubre, Cuenca 010150, Ecuador; jose.pesantez@ucuenca.edu.ec; 4Animal Reproduction Department, SGIT (INIA), Avda. Puerta de Hierro s/n, 28040 Madrid, Spain; andelash@inia.es (A.H.-M.); astiz.susana@inia.es (S.A.); 5Medicine and Surgery Department, Veterinary Faculty, Complutense University of Madrid, 28040 Madrid, Spain; juanvi@ucm.es

**Keywords:** conception rate, ovarian physiology, FTAI, Co-synch

## Abstract

**Simple Summary:**

Reproductive biotechnologies in beef farms are key issues, such as artificial insemination, fixed-time artificial insemination (FTAI), embryo transfer, and ovum pick up, among others. Artificial insemination takes the first place with these available tools. Thus, science keeps improving hormonal treatments and procedures to optimize pregnancies after insemination. A synchronization protocol commonly used in beef heifers is the 5d Co-synch protocol, which fits better in terms of ovarian synchronization and resulting conception rate. We propose a modification of this protocol, which consists of delaying by 24 h the removal of the intravaginal progesterone device, to decrease the percentage of heifers showing estrus before the date of FTAI. Moreover, this modification will make easier the workload when inseminating herds with heifers and cows at the same time. Based on our results, the protocol 5d Co-synch with the delayed removal of the intravaginal progesterone device fits as well as the original protocol in terms of synchronization and conception rate. Therefore, it can be implemented in the heifers of beef cattle commercial farms.

**Abstract:**

In beef herds, increasing animal welfare, improving reproductive performance and easing animal management are key goals in farm economics. We explored whether delaying the removal of the intravaginal progesterone device by 24 h in heifers synchronized with a 5d Co-synch 72-h protocol could improve reproductive efficiency of fixed-time artificial insemination (FTAI). In experiment 1, we examined the total synchronization rate (TSR) in cycling Holstein heifers. Heifers (13.4 ± 0.69 mo.) were randomly assigned to the standard 5d Co-synch 56-h protocol (5dCo56; *n* = 10), 5d Co-synch 72-h (5dCo72; *n* = 17), or the modified 5d Co-synch 72-h protocol, in which removal of the progesterone device was delayed by 24 h (6dCo48; *n* = 19). In experiment 2, 309 cycling beef heifers on 18 commercial farms were subjected to the 5d Co-synch 72-h or 6-d Co-synch 48-h protocol and conception rate (CR) studied. In experiment 1, the three protocols led no differences on TSRs of 80.0% (5dCo56), 88.2% (5dCo72), and 89.5% (6dCo48). In experiment 2, the CR from the beef heifers, observed during two consecutive reproductive seasons did not differ: 59.7% for 5dCo72 and 62.0% for 6dCo48 (*p* = 0.907). Therefore, delaying removal by 24 h provides satisfactory results without reducing reproductive efficiency of heifers.

## 1. Introduction

The first protocol to synchronize ovulation for fixed-time artificial insemination (FTAI) based on gonadotropin-releasing hormone (GnRH) was reported in 1995 [1]. Since then, several modifications have been developed, including such protocols as G6G [2], double-ovsynch [3], presynch-ovsynch [4], five-day ovsynch [5,6], seven-day Co-synch [7], five-day Co-synch [8], and others [9,10]. Research on fixed-time artificial insemination has traditionally focused on dairy cattle, but some protocols and data are available for beef cattle [10]. In fact, certain FTAI protocols have been developed explicitly for beef cattle, such as the five- and seven-day Co-synch protocols plus an intravaginal progesterone (P4) device (IPD) [10].

In GnRH-based protocols, the initial GnRH is intended to induce LH release and ovulation, with emergence of a new follicular wave approximately 2 d later and luteinization of the follicle present at administration guaranteeing an appropriate level of progesterone during follicle growth. The 5d Co-synch protocol, in which FTAI is performed 72 h after the first prostaglandin administration, shortens the follicular dominance by two days and lengthens the proestrus phase by 16–24 h, ultimately leading to higher mean conception rate (CR) [8,11,12,13,14].

Some protocols administer GnRH on Day 0 to enhance subsequent response to GnRH on Day 8 in beef heifers [15] and dairy heifers [16], increasing CR [17]. Some studies suggest that GnRH on Day 0 provides no reproductive benefits in dairy or beef heifers [5,18] and may even induce pregnancy loss in acyclic heifers [19]. However, an FTAI protocol in which cows received prostaglandin three days before the protocol began and then were artificially inseminated at 66 h after IPD removal led to a higher percentage of cows showing a new follicular wave after the initial GnRH as well as higher conception rate than the classical 5d Co-synch 72-h protocol [20]. Such pre-synchronization has been associated with higher CR in dairy heifers [21], higher fertility rate after FTAI in multiparous beef cattle [20], and greater ovarian synchronization in beef heifers [22]. In beef heifers, however, another study did not observe different CRs between the 5d Co-synch 72-h protocol and a prostaglandin 6d-IPD protocol, consisting of prostaglandin (25 mg) on Day −9, GnRH (100 μg) and IPD insertion on Day −6, PG and IPD removal on Day 0, and FTAI at 66 h after IPD removal [23]. Unlike the prostaglandin 6d-IPD protocol, the modified protocol in the present study did not include pre-synchronization with prostaglandin.

In contrast, early work [24] suggested that the 5d Co-synch 56-h (5dCo56) protocol was more appropriate for beef heifers, probably because following P4 device removal, heifers tend to display estrus earlier than cows. However, another study [25] reported higher conception rate in dairy heifers if they received the final GnRH administration concurrent with Artificial Insemination (AI) at 72 h after PGF than if they received it at 16 h before AI. Still another study found no fertility difference between beef heifers who received FTAI at 66 or 72 h after GnRH [19].

A shortened proestrus phase should not affect heifer synchronization or CR, as long as prostaglandin administration is maintained on Days 5 and 6 of the protocol. In fact, the serum concentration of progesterone induced by an intravaginal device is significantly lower than that induced by a natural CL [26]. Prostaglandin administration on Days 5 and 6 promotes corpus luteum (CL) regression [16] and increases CR in dairy and beef heifers [27,28]. While some studies question the usefulness of the second prostaglandin dose in heifers [18,19,29], being indispensable for FTAI programs in multiparous cows [30,31].

A preliminary study [32] found no difference in CR in dairy heifers with a conventional 5d Co-synch 72-h protocol or a modified protocol (Day 0: GnRH + IPD; Day 5: PGF; Day 6: PGF + IPD removal; Day 8: GnRH + FTAI). They described that delaying IPD removal from dairy heifers by 24 h during the 5d Co-synch protocol avoided the demonstration of estrus earlier than 16 h prior to FTAI. However, that study did not examine whether the modified protocol altered the ovarian dynamics compared to the 72-h protocol, nor did it compare total synchronization rates (TSR). The administration of eCG promoting follicular growth in the proestrus stage is another possibility to enhance results, even in cycling animals, as described by other authors [33]. Thus, the optimal window when ovulation should be induced with GnRH relative to FTAI in heifers subjected to the 5d Co-synch protocol warrants further research. Indeed, modification of existing Co-synch protocols may increase the versatility of FTAI programs on beef farms.

New hormonal procedures could increase synchronization and conception rates, together with other strategies, such as nutritional ones [34], body condition score improvement [35], and avoiding stress while handling [36].

The modified protocol, previously proposed [32], henceforth referred to as “6dCo48”, consisted of IPD removal delayed by 24 h relative to the conventional 5dCo72 protocol. We compared these two protocols in terms of ovarian synchronization and conception rates, with the aim at testing its value in the field, in beef heifers. Therefore, the objective of experiment 1 was to examine whether the 6dCo48 protocol would provide an adequate TSR compared to 5dCo72 and 5dCo56 protocols in dairy heifers, based on analysis of follicular growth, luteolysis, and ovulation. Dairy heifers were chosen (easier to obtain an adequate sample of animals homogeneous in age, size, weight and ovarian stage, under a controlled environment) to check that this modification did not significantly worsen the ovarian response nor synchronization rate. The objective of experiment 2 was to compare CR of beef heifers in commercial beef herds after FTAI using the 6dCo48 or 5dCo72 protocols (the two protocols with the best synchronization rate in experiment 1). Our hypothesis was that delaying by 24 h the removal of the intravaginal progesterone device within a 5dCo72 protocol in heifers would not negatively affect conception rates after FTAI and might improve the synchronization rate of heifers, making it an additional synchronization protocol available to beef farmers.

## 2. Material and Methods

### 2.1. Experiment 1: Synchronization Study

Experiment 1 was conducted in a small sample of Holstein heifers to ensure that the new protocol (6dCo48) would not induce a worse synchronization rate than conventional protocols.

Holstein heifers on a single dairy farm in Toledo, in central Spain, were housed in free-stall barns and fed a total mixed ration adapted to their daily requirements [34], with ad libitum access to feed and water. A total of 46 dairy heifers were included, 29 during a first replicate (March 2017) and 17 during a second replicate (April 2017). Two replicates with different heifers were performed due to the few heifers’ availability at once. To be included in the study, animals had to be in adequate health; show a body condition score (defined below) ≥2 and <3.5; lack a history of artificial insemination; and lack anoestrus. Animals needed to present P4 concentrations higher than 1 ng/mL in one or both samples taken seven days apart previous to Day 0 of the study. In addition, initial ovarian cyclicity had to be observed by transrectal ultrasonography on Days −7 and 0 (see Section 2.2.). Of the 51 heifers screened for inclusion, five were excluded because they showed no CL on at least one day, and subsequent P4 assay showed concentrations below 1 ng/mL.

On Day 0 of the protocols, body condition score (BCS) on a scale from 1 (cachectic) to 5 (extremely obese) [35] and stress score (SS) were determined. SS was measured on a scale from 1 to 5 [37] when animals were in the chute, where a score of 1 meant that the animal was calm, without unexpected, sudden movements; 2, the animal was slightly restless; 3, the animal was squirming and occasionally shaking; 4, the animal moved continuously with very vigorous movements that shook the chute; and 5, the animal was rearing, twisting the body, and struggling violently [37].

In the first replicate (March 2017), a total of 29 dairy heifers were assessed for age, BCS, and SS (Table 1) and randomly assigned to undergo the 5dCo72 protocol (*n* = 9), 5dCo56 protocol (*n* = 10), or 6dCo48 protocol (*n* = 10). All protocols involved the insertion of an intravaginal progesterone device. In the second replicate (April 2017), dairy heifers (Table 1) were randomly assigned to undergo our 6dCo48 protocol (*n* = 9) or 5dCo72 protocol (*n* = 8), since the 5dCo72 protocol gave as good TSR as the 5dCo56 protocol in the first replicate (see Results).

The protocol 5dCo72 was performed as described [8]. Briefly, it consisted of intramuscular administration of 100 μg GnRH (Cystoreline^®^ CEVA Santé Animale SA, Libourne, France) and insertion of a 1.55-g IPD (PRID-delta^®^, CEVA) on Day 0. On Day 55, 25 mg dinoprost (Enzaprost T^®^, CEVA) and 500 I.U. eCG (Syncrostim^®^ 500 I.U., CEVA) were administered intramuscularly, and the progesterone device was removed. The 500IU eCG dosage was decided based on drug management guidance for Syncrostim^®^ (SPC product: F-DVM-01-03). This dosage has been demonstrated to be safe and not associated with increased rate of multiple ovulation [33]. On Day 6, 25 mg prostaglandin was administered intramuscularly. On Day 8, 72 h after removal of the intravaginal progesterone device, FTAI was performed and 100 μg GnRH was administered intramuscularly. The 5dCo56 protocol was identical to 5dCo72, except that FTAI and GnRH treatment were performed at 56 h after removal of the progesterone device. The 6dCo48 protocol was identical to 5dCo72, except that the progesterone device was removed on Day 6, simultaneously with the second prostaglandin administration (Figure 1).

Unfortunately, CR after FTAI could not be analyzed, because heifers received conventional or sex-sorted semen from different bulls based on farmer assessments. This may have confounded analyses of CR. Nevertheless, we considered the TSR results to be sufficiently reliable to proceed to a field test of the modified protocol in beef heifers in experiment 2 of the study.

### 2.2. Ultrasonography and Blood Analyses

Transrectal ultrasonography (US) was performed by the same experienced veterinarian on every heifer using a SIUI CTS 800^®^ system (Guangdong, China) with 7.0-MHz transrectal transducer on Days −7, 0, 5, 7.5, and 15. A detailed ovarian analysis was performed in order to link hormonal results to ovarian structure and function. Ovulation was considered to have occurred after the first GnRH if a CL was observed on Day 5 in the ovary where a follicle (diameter > 8 mm) was previously located. Ovulation was considered to have occurred after the second GnRH if on Day 15 there was a CL where the dominant follicle (diameter > 8 mm) was located. Luteolysis was considered to have occurred after the second prostaglandin if on Day 7.5 there was no longer a CL where it had previously been detected. The largest follicle diameter (in mm) was measured on Days 5 and 7.5, and an ovarian map was drawn (from Day 7 to Day 15) in order to study ovarian dynamics.

To explore whether P4 values on D0 influenced synchronization efficacy, we categorized initial P4 concentration on D0 as low (<0.6 ng/mL), intermediate (0.6–7 ng/mL), or high (≥ 7ng/mL). These categories were based on a study in dairy cows [38] that reported higher CR with intermediate P4 concentrations of 0.5–6 ng/mL on Day 0. Attending the metabolic differences between cattle and heifers, the cut-off values were changed in the current study. Moreover, the classical cut-off of <1 or >1 ng/mL P4 was also applied [30] to double check whether P4 values could interfere on the Co-synch modification proposed.

An active CL produces >1 ng/mL P4, and its diameter ranges between 21 and 26 mm [39]. Heifers were considered to be synchronized if they had a CL and P4 > 1 ng/mL on Day 5, underwent luteolysis on Day 7.5, and had a CL and P4 > 1 ng/mL on Day 15. The total synchronization rate (TSR) was calculated based on the number of animals synchronized on Days 5, 7.5, and 15.

Blood was sampled from the coccygeal vein into 4-mL EDTA K2 vacutainer tubes (Fisher Scientific, Pittsburgh, PA, USA), immediately centrifuged at 4500 g for 15 min, and the plasma was transferred to a fresh tube and stored at −80 °C until progesterone determination. Plasma progesterone concentrations were measured in a single analysis using an enzyme immunoassay kit (Demeditec Diagnostics, Kiel-Wellsee, Germany) as described [40]. Assay sensitivity was 0.045 ng/mL and the manufacturer-specified intra-assay variation coefficient was 5%.

### 2.3. Experiment 2: Field Study

A total of 309 beef heifers from 18 commercial beef farms in central and south Spain were included in the study. The animals grazed on pastures and were fed once daily with a complete fodder adapted to their requirements [41] and with ad libitum access to water. Data were collected during two consecutive reproductive seasons, from 187 animals during autumn–spring (2017–2018) and from 122 animals during autumn–spring (2018–2019). Of all 309 animals, 137 were crossbreds, and the remaining 172 heifers were full-bred, comprising 106 Limousine, 55 Charolaise, and 11 Spanish Black Iberian Avileña.

Inclusion criteria for farms were an adequate farm health program, routine SS assessment at the beginning of the reproductive season, and a nutritional program with supplementation when recommended. Individual inclusion criteria for heifers were no prior insemination, nulliparity, age ≥ 17.5 and < 25 months, BCS ≥ 2.5 and ≤ 3.5, and presence of a CL at the beginning of the FTAI protocol. These parameters were measured on Day 0, as described for experiment 1.

Heifers within each farm were randomly submitted to one of two protocols, 5dCo72 or 6dCo48 (Table 2). First, FTAI was performed by two experienced veterinarians using commercial frozen semen from 41 bulls. Pregnancy was diagnosed by US on Day 30–45 after insemination and the conception rate (CR) was calculated. A subset of 122 heifers (39.50%) was randomly selected for blood sampling and P4 assay (see Section 2.2) at the beginning of the 5d Co-synch. This assay allowed us to analyze whether initial P4 affected rates of synchronization.

### 2.4. Statistical Analyses

All data were analyzed using SPSS^®^ 25 (IBM, Armonk, NY, USA). Probability values less than or equal to 0.05 were considered significant, and those between 0.05 and 0.10 were considered trends. All data were reported as mean (percentage) or as mean ± SD. Inter-group differences were assessed for significance using the chi-squared and Student’s t test when data were normally distributed or using non-parametric analyses (Kruskal–Wallis) when data were skewed. Results from both phases of the study were analyzed using logistic regression that included farm as a fixed factor and several possible confounding factors in a stepwise forward method based on the Wald statistic criterion *p* > 0.10. In experiment 2 of the study, data were separately analyzed in two regression models, one with data on all 309 animals, and the other with data on 122 animals with the P4 information.

## 3. Results

### 3.1. Experiment 1

Replicates were clustered by introducing the factor “replicate” in the regression model. Percentages of heifers synchronized at the different time points in the protocol and totally synchronized are summarized in Table 3. TSR did not vary significantly with any of the categorical factors included in the study, including synchronization protocol.

The age on D0 was not different between totally synchronized heifers (13.5 ± 0.89 months) and other heifers (13.6 ± 0.65 mo.; *p* = 0.699). Similarly, the size of the largest follicle on Day 7.5 was not different between totally synchronized heifers (12.8 ± 2.29 mm) and not totally synchronized heifers (12.8 ± 1.16 mm; *p* = 0.968). The P4 values at the time points D0, D5, and D7.5 did not influence the probability of synchronization on the subsequent day where synchronization was evaluated, nor the probability of being totally synchronized at the end of the protocol. The exception was P4 on Day 7.5: each 1-ng/mL P4 increase on Day 7.5 was associated with significantly lower probability of total synchronization at the end of the protocol (OR 0.02, 95% CI 0.001-0.416; *p* = 0.012).

### 3.2. Experiment 2

Pregnant heifers after FTAI (*n* = 188/305) were 21.7 ± 3.77 months old, and non-pregnant ones (*n* = 121/305) were 21.7 ± 3.32 months old (*p* = 0.922). Mean BCS was 3.15 ± 0.478 for pregnant animals, and 3.22 ± 0.503 for non-pregnant ones (*p* = 0.277). Similarly, SS was 2.03 ± 0.701 in pregnant heifers and 2.13 ± 0.763 in non-pregnant ones (*p* = 0.235). Conception rates by group, breed, AI technician, and P4 concentrations on Day 0 are summarized in Table 4.

Farm and semen did not significantly influence conception rate (*p* = 0.907 and 0.329, respectively) in the regression model. Similarly, season (*p* = 0.924), interaction between age and protocol (*p* = 0.746), and interaction between AI technician and protocol (*p* = 0.706) did not significantly affect conception rate.

## 4. Discussion

This study comprises a 5dGnRH-based Co-synch protocol without presynchronization but with GnRH on Day 0 and with prostaglandin administration on Days 5 and 6. The proposed 6dCo48 protocol, a modified version of the conventional 5dCo72 protocol, achieved an ovarian synchronization rate of 89.5% and CR of 62.0%, which were not different from those obtained with the conventional protocol. Thus, the 6dCo48 may be an alternative protocol to optimize the reproductive management of beef heifers subjected to FTAI, enhancing flexibility when considering implementing FTAI strategies in beef cattle farms, without worsening meaningfully reproductive efficiency.

If farmers decide to implement different protocols for cows (5dCo72h) and heifers (5dCo56h), artificial insemination of a single herd must occur in two steps, once in heifers and 16 h later in cows. With the protocol described here, in contrast, beef heifers and multiparous cows can be synchronized and inseminated at the same time. Moreover, as hypothesized, our modification of the conventional 5dCo-synch protocols did not clinically worsen reproductive outcomes in heifers, probably because we maintained prostaglandin administration on Day 5. This mitigated the effect of proestrus shortening on reproductive efficiency. Although the sample size of the study may be a limitation in statistical power, a meaningful difference of 15% CR could have been detected.

A total synchronization rate of 80–90% was observed in the current work with the three hormonal protocols tested in dairy heifers, similarly to previous studies [19,25]. This phase of the study was performed in Holstein heifers, because it was easier to obtain an adequate sample of animals homogeneous in age, size, weight, and ovarian stage that could be maintained in the same controlled environment within a proper handling facility. Moreover, dairy heifers show a comparable ovarian physiology to beef heifers [20,21]. Likewise, a conception rate around 60% was obtained in beef heifers, comparable to that achieved in other studies with the protocol Co-synch 72 and 56 h [9,10,13,14,16,18,24,27,28,29,30,31].

Nevertheless, we cannot exclude that the modified protocol may be associated with higher risk that a heifer has an old oocyte when inseminated [8], resulting in increased pregnancy loss.

In experiment 1, mean age of the 46 dairy heifers in the first part of the study was 13.5 ± 0.86 months, and did not differ between synchronized and other animals. Similarly, the age of beef heifers in the second part of this study did not influence the probability of pregnancy after FTAI. For adequately developed beef heifers, reaching puberty before FTAI may be even more important for fertility than age [23]. Age may not have influenced conception rate in our study because the heifers (21.7 ± 3.54 months old) had reached puberty and had adequate body condition. Since these two factors are the main determinants of fertility in heifers [42,43], they may also explain the lack of differences on TSR between our experimental groups. We also cannot exclude that the lack of a significant difference in TSR is an artifact of the relatively small samples in our study.

Body condition score did not affect the TSR in dairy heifers, which is not surprising given that scores were in the physiological range between 2.5 and 3.5. In contrast, cachectic or extremely obese body condition reduces reproductive performance in cattle [44,45,46]. As in dairy heifers, BCS in our beef heifers also did not affect CR, independently of synchronization protocol. The best reproductive results can be achieved in beef herds [47] with an appropriate nutrition plan [48,49,50] that allows them to achieve a balanced BCS of 2.5–3 at the beginning of the reproductive season. This balanced BCS in heifers is associated with appropriate hormonal regulation [47,50].

Controlling stress reduces cortisol release [51] and ensures cyclical secretion of luteinizing hormone [52,53], improving reproductive efficiency [54,55]. Our dairy heifers showed an SS of 1.71 ± 0.714, consistent with low stress and appropriate management [56]. This may help explain why SS did not influence TSR. SS was similarly low (around 2) in pregnant and non-pregnant beef heifers, which may reflect that beef farms in our study had already implemented stress-reducing measures at the chute. Studies have suggested that SS < 3 indicates good animal temperament [54,55,56,57]. The uniformly low SS in our animals may help explain why conception rate did not differ significantly with beef breed or with the interaction between breed and FTAI protocol. Our results suggest that although CR can depend on temperament in certain breeds [54,55,57], animal temperament in our study was modulated mainly by management.

We found no significant difference in synchronization rates between animals stratified by P4 concentrations on Day 0, using a stratification approach like previous work [38]. That previous work suggested that initial P4 in dairy cows undergoing FTAI should range between 0.5 and 6 ng/mL at the first GnRH administration [38]. When we categorized our animals by initial P4 concentration, we broadened the definition of “intermediate” concentrations to consider that heifers show higher P4 concentrations than dairy cows [58]. We did find a tendency toward higher TSR in dairy heifers with intermediate P4 concentrations on Day 0, but this may not be real, since only one heifer in experiment 1 was in the estrus stage of its cycle on Day 0, and more than 50% of animals showed initial P4 values >7 ng/mL. Our failure to observe a robust dependence of TSR or CR on initial P4 concentration may reflect differences in ovarian physiology between dairy cattle and heifers [59], suggesting that beef heifers can show a strong ovarian response to FTAI protocols independently of initial P4. At the same time, we cannot exclude that the lack of significant variation in TSR or CR reflects the small samples in our study.

Nevertheless, we did find that P4 concentration on Day 7.5, when luteolysis should happen, inversely and significantly affected the rate of synchronization on that day (OR 0.02, 95% CI 0.001–0.416). This may reflect that progesterone inhibits luteinizing hormone release through a negative feedback loop, disrupting ovulation at the end of the synchronization protocol [60].

We found no difference on follicle size around 12.8 ± 2.29 mm before AI across all protocols, also, without differences to the 13.4 ± 0.3 mm reported for beef heifers under the 7-d and 5d Co-synch 72-h protocols [61] and the 11.0 ± 0.5 mm reported for dairy heifers under the 5d Co-synch 72-h protocol with PGF2α presynchronization at two days prior to IPD insertion [21]. Therefore, delaying removal of the intravaginal progesterone device may not meaningfully reduce the TSR.

In conclusion, our modification of the 5d Co-synch protocol for heifers, in which removal of the intravaginal progesterone device is delayed by 24 h, may achieve reproductive results as good as those obtained with 5d Co-synch protocols that stagger insemination of heifers and cows. Therefore, the 6dCo48 protocol seems to be a suitable protocol for beef heifers on commercial farms, which could provide another tool to expand FTAI possibilities. However, due to the limited sample size of this study, additional confirmatory studies are required.

## Figures and Tables

**Figure 1 animals-11-00849-f001:**
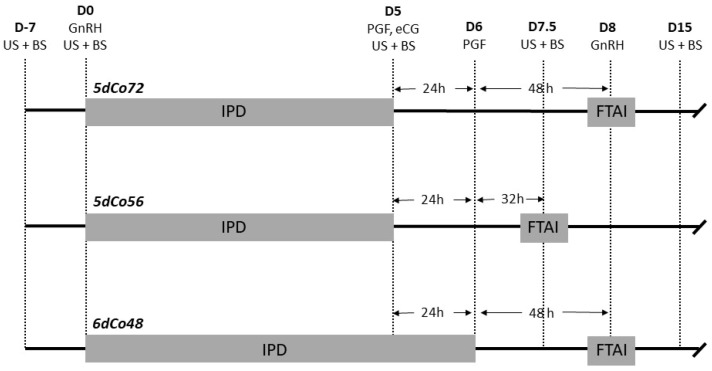
Illustration of treatments and activities during experiment 1: 5dCo72 = 5d Co-synch 72-h protocol with intravaginal progesterone device (IPD); 5dCo56 = 5d Co-synch 56-h protocol with IPD; 6dCo48 = 6-d Co-synch 48-h with IPD; FTAI = Fixed Time Artificial Insemination; D = Day of the study; US = ultrasonography; BS = blood sampling; GnRH = gonadorelin; IPD = intravaginal progesterone device; PGF = prostaglandin; eCG = equine chorionic gonadotropin.

**Table 1 animals-11-00849-t001:** Characteristics of dairy heifers included in experiment 1, by fixed-time AI experimental protocols.

		5dCo72			6dCo48			5dCo56			
Replicates	Factor	*n*	M	SD	*n*	M	SD	*n*	M	SD	*p*-Value *
R1	Age (mo.)	9	13.4	±0.57	10	13.7	±0.97	10	13.8	±0.99	0.235
	BCS (1–5)	9	2.61	±0.22	10	2.65	±0.24	10	2.65	±0.24	0.91
	SS (1–5)	9	1.89	±0.78	10	1.3	±0.48	10	1.5	±0.7	0.192
	P4 (ng/mL)	9	10.57	±8.32	10	10.3	±7.14	10	9.49	±3.81	0.788
R2	Age (mo.)	8	13.3	±0.86	9	13.2	±0.86				0.437
	BCS (1–5)	8	2.88	±0.23	9	2.78	±0.26				0.417
	SS (1–5)	8	1.5	±0.75	9	1.89	±0.78				0.273
	P4 (ng/mL)	8	9.23	±8.42	9	13.65	±5.59				0.29
All	Age (mo.)	17	13.4	±0.69	19	13.5	±0.93	10	13.8	±0.99	0.222
	BCS (1–5)	17	2.74	±0.25	19	2.71	±0.25	10	2.65	±0.24	0.682
	SS (1–5)	17	1.71	±0.77	19	1.58	±0.69	10	1.5	±0.7	0.765
	P4 (ng/mL)	17	9.94	±8.13	19	8.18	±7.12	10	9.49	±3.81	0.55

Abbreviations: M = Mean; SD = Standard Deviation; Age: age at study inclusion (Day 0); BCS = Body Condition Score at Day 0, R = Replicate; scale 1–5; SS = Stress Score (1–5) at Day 0; P4 = plasma progesterone concentration on Day 0; 5dCo72 = 5d Co-synch 72-h protocol with intravaginal progesterone device (IPD); 5dCo56 = 5d Co-synch 56-h protocol with IPD; 6dCo48 = 6-d Co-synch 48-h protocol with IPD. * *p*-value after Kruskal–Wallis analysis.

**Table 2 animals-11-00849-t002:** Characteristics of beef heifers included in experiment 2, by fixed-time AI experimental protocols.

Factor	5dCo72			6dCo48			*p*-Value *
	*n*	Mean	SD	*n*	Mean	SD	
Age (mo.)	159	21.2	±3.76	150	±22.1	±3.36	0.222
BCS (1–5)	159	3.15	±0.42	150	±3.19	±0.55	0.682
SS (1–5)	159	2.09	±0.7	150	±2.05	±0.75	0.55
P4 (ng/mL)	60	5.49	±5.01	62	±6.01	±6.09	0.765

Abbreviations: SD = Standard Deviation; Age: age at study inclusion (Day 0); BCS = Body Condition Score at Day 0, scale 1–5; SS = Stress Score (1–5) at Day 0; P4 = plasma progesterone concentration on Day 0; 5dCo72 = 5d Co-synch 72-h protocol with intravaginal progesterone device (IPD); 6dCo48 = 6-d Co-synch protocol 48-h with IPD. * *p*-value after Kruskal–Wallis analysis.

**Table 3 animals-11-00849-t003:** Summary of synchronization results of dairy heifers in experiment 1 by synchronization days.

Factor	Value	TSR (N/*n*)	*p*-Value *	S-D15 (*n*/N)	*p*-Value *	S-D7.5 (*n*/N)	*p*-Value *
BCS	2.5	85.2% (23/27)	0.593	92.6% (25/27)	0.238	88.9% (24/27)	0.809
	3	89.5% (17/19)		100% (19/19)		89.5% (17/19)	
SS	1	87.5% (21/24)	0.801	95.8% (23/24)	0.818	87.5% (21/24)	0.435
	2	81.3% (13/16)		93.8% (15/16)		87.5% (14/16)	
	3	100% (6/6)		100% (6/6)		100% (6/6)	
Protocol	5dCo72	88.2% (15/17)	0.573	100% (17/17)	0.472	88.2% (15/17)	0.657
	6dCo48	89.5% (17/19)		94.7% (18/19)		89.5% (17/19)	
	5dCo56	80.0% (8/10)		90.0% (9/10)		90.0% (9/10)	
P4-Day 0	<0.6 ng/mL	0.0% (0/1)	0.846	100% (1/1)	0.465	0.0% (0/1)	0.161
	>0.6 and <7 ng/mL	88.9% (16/18)		100% (18/18)		88.9% (16/18)	
	≥7 ng/mL	85.2% (23/27)		92.6% (25/27)		92.6% (25/27)	

Abbreviations: BCS = Body Condition Score on Day 0 (scale 1–5); SS = Stress Score (1–5) on Day 0; 5dCo72 = 5d Co-synch 72-h protocol with intravaginal progesterone device (IPD); 5dCo56 = 5d Co-synch 56-h protocol with IPD; 6dCo48 = 6-d Co-synch 48-h protocol with IPD. P4-Day 0, plasma progesterone concentration on Day 0; TSR = total synchronization rate, SD-15 = synchronization on Day 15; SD-7.5 = synchronization on Day 7.5. * From logistic regression modelling.

**Table 4 animals-11-00849-t004:** Conception rates of beef heifers in experiment 2 and effect of the different factors analyzed.

Factor	Class	N	CR (*n*/N)	*p*-Value *
Group	5dCo72	159	59.7% (95/159)	0.685
	6dCo48	150	62.0% (93/150)	
Breed	Charolais	55	63.6% (35/55)	0.581
	Limousine	106	63.2% (67/106)	
	Crossbreed	137	56.9% (78/137)	
	Spanish Black Iberian Avileña	11	72.7% (8/11)	
Stress	0	68	61.8% (42/68)	0.237
	1	154	64.3% (99/154)	
	2	84	54.8% (46/84)	
	3	3	33.3% (1/3)	
AI-Tech	1	177	58.2% (103/177)	0.269
	2	132	64.4% (85/132)	
P4-Day 0	<0.6 ng/mL		66.7% (10/15)	0.900
	>0.6 < 7 ng/mL		60.3% (41/68)	
	≥7 ng/mL		61.5% (24/39)	

CR = conception rate; 5dCo72= 5d Co-synch 72-h protocol with IPD; 6dCo48 = 6-d Co-synch 48-h protocol with IPD. AI-Tech: technician who performed artificial insemination; P4-Day 0: progesterone plasma concentration measured on Day 0; * From logistic regression modelling.

## Data Availability

The data presented in this study are available on request from the corresponding author.

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
