# Peer review of "Effects on Synchronization and Reproductive Efficiency of Delaying the Removal of the Intravaginal Progesterone Device by 24 h in the 5d Co-Synch Protocol in Heifers"

_animals, 2021, doi:10.3390/ani11030849_

Round 1

Reviewer 1 Report

The presented paper deals with the effects of a 24 hours delayed remove of intravaginal progesterone device in the 5 days Co-Synch protocol on the total synchronization and conception rates in milk and beef heifers. This study attempts to determine how far the above described modification of the standard Co-Synch protocol may influence the ovarian physiology and some reproductive performance in heifers. In my opinion, this topic is current and in general is within the scope of the journal. However, I have to add, that the studied issue is very narrow and is interesting mainly from practical point of view. The evaluated study is focused on the physiological and reproductive effects only after a small modification of well established hormonal protocol for synchronization of estrus cycle. Then, the final decision, if this manuscript is worth being published in “Animals”, should rest with Editor. In my opinion, the outcomes of this study may be interesting for the readers interested in practical work.

The general study design is satisfactory, methodology is rather acceptable, as well as the results and conclusions are supported by the obtained data. The whole manuscript is rather well-structured and clearly written, due to this the entire paper is easy to follow.

Although, in my opinion this manuscript has required quality, I have some concerns regarding its editorial aspects. I would suggest to take into consideration as follows:

  • the title of this paper should be improved, please give detailed information about the effects studied,
  • some phrases from the “Introduction” (lines 86-91) and “Material and Methods” (lines 106-112) should be moved to Discussion,
  • a limitation of your study is lack of information about a total pregnancy rate after the use of the modified protocol. It is possible, that this protocol may be related to the oocytes aging and in consequence increased embryonal or fetal loss. Please, give this comment in Discussion.
  • Discussion seems to be overlong, sometimes you discuss the well-known issues. Please, try to shorten this chapter.

In conclusion, I recommend this manuscript for publication conditionally (see the remarks above). After Editor’s positive decision, the minor revision is needed.

Author Response

We are thankful with the criticism of the reviewer and have included detailed answers for his/her appreciations. All changes from reviewer 1 are in yellow highlighted in the manuscript.

A point by point response to the reviewer's comments is in the word attached

Reviewer 2 Report

This manuscript presents the results of two experiments aimed at investigating the potential consequences of extending the duration of CIDR treatment in a 5-day CO-Synch + CIDR treatment. Authors argue there may be some relevance to commercial production systems. Unfortunately, neither experiment was designed with enough power to detect a meaningful difference, as the response of interest (synchronization rate, pregnancy per AI) are bivariate (yes/no) responses that would require much larger animal numbers. Unexpectedly, authors fail to detect a significant difference between treatments. However, authors infer that this indicates equivalence of the treatments. This is not true; failure to detect a difference is not the same as demonstration of equivalence. This can be show to be the case be conducting an equivalency or non-inferiority test, or simply by evaluating the relatively wide confidence intervals around the means observed. Because equivalency was hypothesized, this manuscript will require significantly more animal numbers before it would merit publication. 

Author Response

We appreciate this comment from the reviewer. We, scientist are steadily confronted with the conflict between using a small sample size of animals under controlled, experimental situations, or larger samples under commercial circumstances, and in both situations limitations of available animals or validity of the results are present. Larger amounts of inseminations are normally from epidemiological, observational studies, with their own (other) limitations.

We agree with the referee in his/her statements that the current experimental design does not allow us to speak of equality of the groups. However, asssuming a percentage of conception rate of 45%, a confidence level of 95% and the given sample size of 150 animals per group, it is calculated a Confidence interval of 7.9. On the other way, in order to be sure that we could detect a difference of 15% or more (clinically important and meaningful difference) in CR between groups, a sample size of 173 per group would be right with an alfa error of 0.05and a beta error of 0.2; and a power of 0.8.

Moreover, we find examples of several studies on bovine reproduction and similar issues (ovarian dynamics, fixed time artificial insemination protocols), published recently in different journals with similar number of animals included, such as: Lopes et al in September 2020 (56 and 437 animals), Hye et al. in May 2020 (297 animals), Stewart et al. in May 2020 (12 animals), Leonardi et al. in January 2020 (30 and 18 animals)… Therefore, we would like to emphasize that the sample size and experimental design of this study report valid results, contrary to that stated by referee 2.

It is difficult to be completely sure of one statement in fertility in cows after one single study, but we are convinced that our results are worthy to be shared with the Academia, and that they enhance knowledge.

We have changed the wording in several sentences to make clear the point that the reviewer highlight to correct our not exact enough statements and have soften some categorical statements. Lines 303, 306, 311-312, 314-315, 369-370, 381, 383-384, 385-388.

Reviewer 3 Report

Lines

22 first and not fist

48 Introduction : It could be interesting to explain why some protocols have been specifically developped for beef cattle.  Moreover, as you are using eCG, it’s also important to explain the justification of such hormone in the protocol particularly in heifers who are cycled.

56 it’s necessary also to mention and justify the use of one or two PGF2a. Moreover to explain the effect of progesterone on follicular growth  will be appreciated. IPD are especially used for anoestrus animals. What coul be the interest in cycled animals ?

59 what about the possibility to have an accessory corpus luteum or a luteinization of the dominant follicle ?

74 for what reasons some animals don’t show a luteal regression after the first injection of PGF2a ?

97 not only the ovarian response I suppose but also the synchronization rate

116 using the word replicate, the lector can understand that the heifers have been use two times. Could you explain why you have use two groups and why you have separate your observations ?

125 could you explain in the introduction the interest to determine the stress score attending you comment this aspect in discussion?

184 Do you believe that the comparison with dairy cows is normal attending that the catabolism of P4 is different in heifers and cows ?

186 zero and not cero

245 . Similarly, the size of the largest follicle on Day 7.5 was not different between totally synchronized heifers (12.8 ± 2.29 mm) and others (12.8 ± 1.16 mm; P = 0.968) : this sentence is not enough understandable : what’s the difference between synchronized heifers and others heifers ? We can regret that the diameter of the follicle has not been measured at the time of insemination. Do you have an explanation ?

247 The P4 values at different time points in the protocol 247 did not influence the probability of synchronization on the nearest subsequent measure- 248 ment date, nor the probability of being totally synchronized at the end of the protocol : wat do you try to explain. Can you re-write this sentence ?

254 Table 2 : Why the S-D5 has not been presented ?

266 Table 4 Why the SS scores don’t have been taken in account ?

Author Response

We appreciate the important and interesting comments the reviewer. We have included them all throughout the manuscript and they have been highlighted in pink.

A point by point response to the reviewer's comments is in the word attached

Round 2

Reviewer 2 Report

This reviewer stands by the previous review and encourages the authors to add additional replicates prior to preparing a future submission.